## PERSPECTIVE

### RyR cooperativity and mobile buffers: functional clues to the resolution of the cardiac calcium wave problem?

Michael A. Colman

*University of Leeds, Leeds, UK*

Email: M.A.Colman@leeds.ac.uk

Handling Editors: Natalia Trayanova & Eleonora Grandi

The peer review history is available in the Supporting Information section of this article (https://doi.org/10.1113/JP287762#support-information-section).

Spontaneous calcium ($Ca^{2+}$) sparks and propagating $Ca^{2+}$ waves have been closely linked to the cellular mechanisms underlying cardiac arrhythmias. The resulting spontaneous $Ca^{2+}$ transients can activate the sodium–calcium exchanger, generating an inward current that may lead to early or delayed afterdepolarisations that can serve as both triggers and substrate for arrhythmogenesis (Liu et al., 2015). Understanding these phenomena is therefore crucial for elucidation of the mechanisms of arrhythmia.

Spontaneous $Ca^{2+}$ sparks occur in restricted subspaces that mediate $Ca^{2+}$-induced-$Ca^{2+}$-release (CICR) and arise from the spontaneous opening of clusters of ryanodine receptors (RyRs), the channels responsible for $Ca^{2+}$ release. Diffusion of $Ca^{2+}$ between neighbouring clusters provides a mechanism for the nucleation and propagation of spontaneous $Ca^{2+}$ waves throughout the cell. This 'fire-diffuse-fire' mechanism posits a reasonable and simple explanation for the propagation of $Ca^{2+}$ waves. Given this, one could very well then ask: wherein lies the problem?

The issue at the heart of the $Ca^{2+}$ wave problem is that our contemporary computational models of spatial $Ca^{2+}$ handling at the whole-cell scale, based on our best functional and structural knowledge, struggle to reproduce sustained, propagating $Ca^{2+}$ waves under physiological conditions. This suggests that something is missing from our theoretical understanding.

For a $Ca^{2+}$ spark to propagate between clusters, the local concentration elevation at the neighbouring cluster must be sufficient to induce CICR. Yet, it is necessarily not much higher than diastolic levels due to buffering and the impact of diffusion in three dimensions; RyR model parameters that induce CICR at these concentrations often lead to excessive spontaneous sparks and arrhythmogenesis. Models have therefore had to make compromises to solve this problem (Colman et al., 2022), either incorporating 'natural' spatial coupling but relying on large $Ca^{2+}$ transients ($>4$ μM) to propagate $Ca^{2+}$ waves, or by 'artificially' strengthening the spatial coupling through the inclusion of coupled $Ca^{2+}$ subspaces, with the benefit of maintaining physiological $Ca^{2+}$ transients ($\sim$0.5–1 μM). It seems, at least, almost impossible for detailed models to fulfil both normal CICR and sustained $Ca^{2+}$ waves without relying on these coupled subspaces, which lack structural evidence.

It is likely no coincidence that diseased myocytes that are very prone to sustained $Ca^{2+}$ waves have been observed with severe structural remodelling, such as a loss of the transverse and axial tubule system, fragmentation of the arrangement of RyRs within clusters and rearrangement of inter-cluster spacing, and remodelling of other organelles such as mitochondria. For instance, Miragoli et al. (2016) showed that microtubule disorganization displaced mitochondria, creating local structures that could facilitate $Ca^{2+}$ wave propagation. These factors can't, however, explain the observation of $Ca^{2+}$ waves in non-diseased myocytes. Whereas it may seem unnecessary to focus substantial efforts on reproducing $Ca^{2+}$ waves in healthy conditions, given that their incidence is likely too low to have any impact on tissue dynamics, it is nevertheless imperative that we can construct models that reproduce simultaneously the range of normal physiological behaviours, such that we can have confidence in the translation of those models to disease conditions.

Recognising the importance of this fundamental challenge, Zhong and Karma (2024), develop a highly sophisticated model of spatial $Ca^{2+}$ handling at the whole-cell scale that is applied to explore the impact of various functional parameters on CICR and $Ca^{2+}$ wave dynamics. A primary focus of their study is the cooperativity of the RyRs within a single cluster. Similar to previous work showing that increased cooperativity among subunits of a single RyR channel enabled stable closed states at diastolic $Ca^{2+}$ levels and robust CICR sensitivity (Greene et al., 2023), the authors found that higher cooperativity supports $Ca^{2+}$ wave propagation without inducing unphysiologically large $Ca^{2+}$ spark rates: the model enabled sustained $Ca^{2+}$ waves under parameters that gave a $Ca^{2+}$ transient magnitude closer to that expected ($\sim$2 μM *vs.* 0.5–1 μM, rather than $\sim$4+ μM of previous studies). It is notable, however, that in all conditions with a $Ca^{2+}$ transient during normal pacing that is closer to 1 μM, no waves were sustained.

Other important factors emerged from their analysis that may provide further clues to the underlying dynamics that facilitate $Ca^{2+}$ wave propagation. The authors found that the inclusion of mobile ATP buffering was *necessary* to sustain $Ca^{2+}$ waves and delayed afterdepolarisations; this feature may significantly contribute to the improvements seen in their model compared with previous ones. These results perhaps offer intriguing insight into the solution to this problem: our coarse-grained, simplistic approximations of $Ca^{2+}$ diffusion as a continuum may be inadequate to capture the underlying complexity at these spatial scales and within the intricate sub-cellular geometry. Careful consideration of the local pathways that $Ca^{2+}$ ions may take between adjacent RyR clusters may be required to fully explain $Ca^{2+}$ wave propagation in a range of conditions.

Undoubtedly, multiple factors including RyR cooperativity, mobile buffers and structural relationships such as RyR arrangement and inter-cluster coupling, will work together to explain experimental observations. The interplay between these multiple factors, and how each is modulated in disease to provide a highly arrhythmogenic substrate, remains to be fully elucidated. This study by Zhong and Karma, which explores multiple factors in much greater detail than is summarized here, represents an exciting step towards resolving a core issue in our understanding of (patho)physiological cellular and tissue electrical dynamics.

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

## Additional information

### Competing interests

None declared.

### Author contributions

Sole author.

### Funding

UKRI | Medical Research Council (MRC): Michael A Colman, MR/V010050/1

## Keywords

arrhythmia mechanisms, cardiac calcium handling, computational modelling

## Supporting information

Additional supporting information can be found online in the Supporting Information section at the end of the HTML view of the article. Supporting information files available:

**Peer Review History**

