## [Peer Review History · The Journal of Physiology]

RyR cooperativity and mobile buffers: functional clues to the resolution of the cardiac calcium wave problem?

Michael A Colman
DOI: 10.1113/JP287762

Corresponding author(s): Michael Colman (M.A.Colman@leeds.ac.uk)

Review Timeline:

Submission Date:	07-Oct-2024
Editorial Decision:	14-Oct-2024
Revision Received:	17-Oct-2024
Accepted:	01-Nov-2024

Senior Editor: Natalia Trayanova

Reviewing Editor: Eleonora Grandi

Transaction Report:

Dear Dr Colman,

Re: JP-P-2024-287762 "RyR cooperativity and mobile buffers: functional clues to the resolution of the cardiac calcium wave problem?" by Michael A Colman

Thank you for submitting your manuscript to The Journal of Physiology. It has been assessed by a Reviewing Editor and by an expert referee and we are pleased to tell you that it is acceptable for publication following satisfactory revision.

Please address all the points raised and incorporate all requested revisions or explain in your Response to Referees why a change has not been made. We hope you will find the comments helpful and that you will be able to return your revised manuscript within 4weeks. If you require longer than this, please contact journal staff: jp@physoc.org.

REVISION CHECKLIST:

- 'Potential Cover Art' for consideration as the issue's cover image
- Appropriate Supporting Information (Video, audio or data set: see https://jp.msubmit.net/cgi-bin/main.plex?form_type=display_requirements#supporting_information)

form_type=display_requirements#supp).

We look forward to receiving your revised submission.

Yours sincerely,

Natalia Trayanova
Senior Editor
The Journal of Physiology

REQUIRED ITEMS

EDITOR COMMENTS

Reviewing Editor:

Thank you for an excellent editorial. The reviewer raises a suggestion for clarification.

REFEREE COMMENTS

Referee #1:

This is an excellent summary of the work of Zhong and Karma that achieves to communicate the main findings to a broad audience without getting too far down into the weeds of technical details that abound in this work. One place where I think that the text could perhaps be clarified further in the first part of the sentence:

"Whereas it was challenging to separate the direct impact of RyR cooperativity from the secondary effects of Ca²⁺ load and magnitude of Ca²⁺ release,..."

I find the message ambiguous in that "whereas it was challenging to separate..." can be interpreted to mean that the results were not able to separate the direct impact of RyR cooperativity and SR load.

While it is true that inferring causality can be tricky in a complex system such as a cardiomyocyte with many interacting parts, Fig. 8 of Zhong-Karma seems to clearly show that elevated SR load increases with RyR cooperativity (panels Aa and Bb) that reduces the frequency of CRU opening at small diastolic cleft Ca concentration (panel D). So, from this figure, the causal chain appears to be that increased RyR cooperativity suppresses excessive Ca sparks that in turn sustains SR load above the threshold necessary for Ca wave propagation. This causal chain is in qualitative agreement with the classic experiment of Eisner et al showing that both increased RyR leakiness and elevated SR load are necessary for Ca wave propagation (e.g. combination of caffeine and ISO). The present work clarifies that increased leakiness without sufficient RyR cooperativity does not suffice to maintain SR load above this threshold.

This is just an optional suggestion for clarification.

END OF COMMENTS

I thank the editor and reviewer for their rapid and constructive comments.

Comment: "Whereas it was challenging to separate the direct impact of RyR cooperativity from the secondary effects of Ca²⁺ load and magnitude of Ca²⁺ release,..."

I find the message ambiguous in that "whereas it was challenging to separate..." can be interpreted to mean that the results were not able to separate the direct impact of RyR cooperativity and SR load.

While it is true that inferring causality can be tricky in a complex system such as a cardiomyocyte with many interacting parts, Fig. 8 of Zhong-Karma seems to clearly show that elevated SR load increases with RyR cooperativity (panels Aa and Bb) that reduces the frequency of CRU opening at small diastolic cleft Ca concentration (panel D). So, from this figure, the causal chain appears to be that increased RyR cooperativity suppresses excessive Ca sparks that in turn sustains SR load above the threshold necessary for Ca wave propagation. This causal chain is in qualitative agreement with the classic experiment of Eisner et al showing that both increased RyR leakiness and elevated SR load are necessary for Ca wave propagation (e.g. combination of caffeine and ISO). The present work clarifies that increased leakiness without sufficient RyR cooperativity does not suffice to maintain SR load above this threshold.

Response: I experimented a little with different approaches to modification of this sentence to more accurately reflect the original study, as highlighted by the reviewer. Due to the nature of this short review and it already being at the word limit, I found that the cleanest and clearest way to address this inaccuracy/ambiguity in the text was simply to remove that statement. The section now reads:

"...the authors found that higher cooperativity supports Ca²⁺ wave propagation without inducing unphysiologically large Ca²⁺ spark rates: The model enabled sustained Ca²⁺ waves under parameters that gave..."

I am happy to modify this further if the reviewer or editor feels this is necessary.

Dear Associate Professor Colman,

Re: JP-P-2024-287762R1 "RyR cooperativity and mobile buffers: functional clues to the resolution of the cardiac calcium wave problem?" by Michael A Colman

We are pleased to tell you that your paper has been accepted for publication in The Journal of Physiology.

Yours sincerely,

Natalia Trayanova
Senior Editor
The Journal of Physiology

If you would like to receive our 'Research Roundup', a monthly newsletter highlighting the cutting-edge research published in The Physiological Society's family of journals (The Journal of Physiology, Experimental Physiology, Physiological Reports, The Journal of Nutritional Physiology, and The Journal of Precision Medicine: Health and Disease), please click this link, fill in your name and email address and select 'Research Roundup':

<https://www.physoc.org/journals-and-media/membernews>

- You can help your research get the attention it deserves! Check out Wiley's free Promotion Guide for best-practice recommendations for promoting your work at: www.wileyauthors.com/eeo/guide. You can learn more about Wiley Editing Services which offers professional video, design, and writing services to create shareable video abstracts, infographics, conference posters, lay summaries, and research news stories for your research at: www.wileyauthors.com/eeo/promotion.

The Corresponding Author will receive an email from Wiley with details on how to register or log-in to Wiley Authors Services where you will be able to place an order

EDITOR COMMENTS

Reviewing Editor:

Thanks for a nice perspective article

REFEREE COMMENTS

Referee #1:

The author has satisfactorily addressed the reviewer's comments.

1st Confidential Review

17-Oct-2024